# Analog Memristive Characteristics of Square Shaped Lanthanum Oxide Nanoplates Layered Device

**DOI:** 10.3390/nano11020441

**Published:** 2021-02-09

**Authors:** Wonkyu Kang, Kyoungmin Woo, Hyon Bin Na, Chi Jung Kang, Tae-Sik Yoon, Kyung Min Kim, Hyun Ho Lee

**Affiliations:** 1Department of Chemical Engineering, Myongji University, Yongin-Si 17058, Korea; kangwk7@naver.com (W.K.); mywkm1@naver.com (K.W.); hyonbin@mju.ac.kr (H.B.N.); 2Department of Physics, Myongji University, Yongin-Si 17058, Korea; cjkang@mju.ac.kr; 3Department of Materials Sciences and Engineering, Myongji University, Yongin-Si 17058, Korea; tsyoon@mju.ac.kr; 4Department of Materials Sciences and Engineering, KAIST, Daejeon-Si 34141, Korea; km.kim@kaist.ac.kr

**Keywords:** analog resistive switching, square shape, lanthanum oxide, neuromorphic device

## Abstract

Square-shaped or rectangular nanoparticles (NPs) of lanthanum oxide (LaO_x_) were synthesized and layered by convective self-assembly to demonstrate an analog memristive device in this study. Along with non-volatile analog memory effect, selection diode property could be co-existent without any implementation of heterogeneous multiple stacks with ~1 μm thick LaO_x_ NPs layer. Current–voltage (I–V) behavior of the LaO_x_ NPs resistive switching (RS) device has shown an evolved current level with memristive behavior and additional rectification functionality with threshold voltage. The concurrent memristor and diode type selector characteristics were examined with electrical stimuli or spikes for the duration of 10–50 ms pulse biases. The pulsed spike increased current levels at a read voltage of +0.2 V sequentially along with ±7 V biases, which have emulated neuromorphic operation of long-term potentiation (LTP). This study can open a new application of rare-earth LaO_x_ NPs as a component of neuromorphic synaptic device.

## 1. Introduction

Digital-type resistive switching, (RS) showing bistability between high resistance state (HRS) and low resistance states (LRS) has been continuously investigated with a substantial resistance ratio gap in a metal oxide layer for various non-volatile memory and electronic switch devices [1]. Thus, the digital RS phenomena have been observed and developed in numerous dielectric metal oxides. To fabricate the RS active layer, the metal oxide film has been typically deposited by various methods such as sputtering, atomic layer deposition (ALD), and chemical vapor deposition (CVD). Meanwhile, an analog-type RS device with sequential modulation of history adaptive resistance changes has recently attracted extensive interest for applications such as analog non-volatile memory, programmable analog circuits, and neuromorphic functionality [2,3,4,5,6,7]. Examples of the analog memory behaviors in metal oxide nanoparticles (NPs) such as iron oxide (Fe_2_O_3_) have been successfully demonstrated [2,3]. In addition, there has been a recent start with a new demand to integrate digital and analog RS operations in one integrated system simultaneously for future electronics. Thus, it would be beneficial to provide both digital and analog characteristics with a single material system operating under different conditions, for example, such as thickness [4].

So far, the digital RS device has been widely pursued and reported for a wide range of systems based on metal oxide films or NP layers. However, analog RS with the metal oxide systems has not been routinely achieved [5,6], which have demonstrated analog RS operations through the thin film formed by vacuum-based deposition techniques [7]. Apart from the vacuum-processed formation of the RS active layer, spherical metal oxide NPs have been demonstrated to accomplish the analog RS effects in active layers of Fe_2_O_3_, Pt-Fe_2_O_3_, and NiO NPs stacks. The formation of multiple NP layers was enabled and processed by multiple dip-coatings based on Van der Waals adsorption [2,3,4]. Otherwise, hybrid nanocomposite layers with polymeric matrices or binders such as poly(methyl methacrylate) (PMMA), polystyrene (PS) and polyvinylphenol (PVP) have been pursued for the analog RS device [8,9,10].

Importantly, the role of the NPs as component of the RS active layer can confer other advantages over a principal role of charging trapping element by variation of the NP’s shape. For example, the conventional spherical shape of NPs can be intentionally designed into a non-spherical or non-isotropic shape during the synthesis stage, which could be controlled by varying the process temperature and time [11]. Also, chemically synthesized NPs can be prepared by a solution-based process which is cost effective and applicable to organic or plastic substrates. The spherical NPs layered in RS devices would have isotropic effects under applied bias [2,3,4,5]. Particularly, with respect to the layer formation process, the spherical NPs have the advantage of accomplishing an even and uniform film by the isotropic shape effect [12,13,14].

However, for oxide-based NPs, there have been a limited number of reports about the non-spherical NPs. For example, nanorods or rectangular ZnO NPs have been demonstrated to take advantage of anisotropic shape effects apart from popularity of the ZnO as semiconducting material [15,16,17,18]. Here, non-spherical ZnO nanorods were adopted as electroluminescent structures [15], rectangular ZnO NPs enhanced photocatalytic activity [16,17], and antibiotic activity was exploited [18]. These featured characteristics are believed to have originated mainly from edge effects of the nanorod and rectangular shapes.

Meanwhile, there have been many applications for transition metal oxides (HfO_2_, ZnO, La_2_O_3_ etc.) of high k dielectrics, which have been dedicatedly developed for complementary metal-oxide-silicon cMOS device fabrication [19,20,21]. Especially for rare-earth (RE) species with the transition metal oxides, recent attentions have been raised due to their optical and catalytic capabilities. For example, La_2_O_3_ has been known to have a high k (ε = 30) dielectric constant and strong insulation against electrical break down [22]. Especially, the processes of La_2_O_3_ layer formation have been extensively developed and performed by reactive sputtering of the La target with oxygen gas. Otherwise, ALD using a complicated La precursor such as lanthanum-2,2,6,6-teramethyl-3,5-heptanedione has been extensively developed [22,23]. However, reactive sputtering of La with O_2_ gas should be restricted to its unstable La_2_O_3_ composition. Moreover, for the La precursors used to the ALD process, a high temperature (>500 °C) is required to form a stable film.

In this study, the non-spherical shaped or square-shaped LaO_x_ NPs layer between Au and Pt electrodes (Au/LaO_x__/_Pt) was fabricated and characterized for the non-volatile RS device. Actually, so far, very limited research has been pursued with regard to stoichimetric La_2_O_3_ or non-stoichiometric LaO_x_ targeted to active materials for RS property [22,23]. The main reason for the small number of reports about the La_2_O_3_ based RS device would be the restriction of the high temperature requirement for annealing [22,23]. However, in this study, since chemically stable LaO_x_ NPs were colloidally synthesized prior to device formation and then they were implemented for the active layer of the RS device, fewer difficulties in the fabrication process could be encountered. Particularly, the LaO_x_ NPs were stabilized by ligands, which have influenced unique RS characteristics due to the cooperative effects among individual NPs [24,25]. In addition, so far, there has been no available method to synthesize LaO_x_ NPs in an isotropically spherical shape [15,24]. Spontaneously, the synthesized NPs have been in nanoplate- or rectangle- shaped colloids. The synthesis limitation for the isotropically spherical shape had to result in a ~1 μm thick layer of LaO_x_ NPs for the non-volatile synaptic device. However, the first insight of non-spherical NPs for analog memristive device will be meaningful, which can have an edge or anisotropic effect from the square-shaped LaO_x_ NPs.

## 2. Experimental

### 2.1. LaO_x_ Nanoparticles Preparation and Characterization

To prepare the synthesis solution for LaO_x_ NPs synthesis, 2 mmol of lanthanum (III) acetate hydrate was added to a mixture of 60 mmol of oleyl amine and 18 mmol of oleic acid at room temperature. After degassing the mixture at 90 °C in a vacuum, it was heated up to 320 °C in an N_2_ atmosphere. It was further refluxed at 320 °C for 2 h. After cooling down the resulting solution to room temperature, a pale yellow precipitate was collected by the addition of ethanol and following centrifugation. The precipitate was dispersed in hexane and precipitated again with ethanol to remove unreacted precursor and excess ligands. After a series of the workup procedures, LaO_x_ nanoplate dispersion was dispersed in hexane. Using a high-resolution transmission electron microscopy (HR-TEM, JEM-2100F, Bruker Co. Billerica, MA, USA), the shape of the LaO_x_ nanoplates was analyzed.

### 2.2. Fabrication of Au/LaO_x_ NPs/Pt Device

The metal–insulator–metal (MIM) structure of Au/LaO_x_ NPs/Pt was fabricated on Si substrate as follows. First, on a Si wafer, a 300 nm SiO_2_ layer was thermally oxidized. Then, the bottom Pt electrode layer (150 nm) was deposited onto the SiO_2_ layer by sputtering. The LaO_x_ NPs layer was then formed on the bottom Pt electrode by the repeated dip-coating process. The dip-coating process involved vertically dipping the substrate in the LaO_x_ NPs dispersed in *n*-hexane. The substrate was pulled out at a speed of 0.7 mm/min. After the dip-coating, the substrate was dried in air. In the dip-coating process, the NP layer was formed by Van der Waals adsorption, and the corresponding assembly was formed upon the evaporation of the solvent or by the convective self-assembly of the NPs being driven into the meniscus of the solution at the substrate surface [9]. The NP layer was subsequently annealed at 300 °C for 15 min in air to decompose the surfactants or ligands from the NPs surface. The dip-coating and annealing step were repeated twice in turn, thus, forming a 1 µm thick mixed NP layer. Finally, the Au top electrode with a diameter of 500 µm was deposited by evaporation with a shadow mask.

### 2.3. Electrical and Physical Characterization of the Au/LaO_x_ NPs/Pt Device

The current–voltage (I–V) characteristic of the device was examined by an Agilent 4156C semiconductor parameter analyzer. For the examination of the LTP operation of the LaO_x_ NPs device, an Agilent 41501B pulse generator was used to control the pulse width and interval time. An FE-SEM (field emission scanning electron microscope, SU-70 Hitachi Co., Osaka, Japan) showed the cross-sectional and top image of the device, and FE-SEM (S-4800, Hitachi Co., Osaka, Japan) was also used for the analysis of the LaO_x_ device surface. The TEM sample with a complete device was fabricated by focus ion beam (FIB, NOVA 600 Nanolab, FEI Co., Hillsboro, OR, USA).

## 3. Results and Discussions

Figure 1a shows a high-resolution TEM image which reveals plate shapes of syn-thesized LaOx NPs. The side lengths of the LaOx NPs were measured as about 5 nm–20 nm as shown in Figure 1a. It was clearly found that the NPs were formed in plate or square shapes. Figure 1b shows a schematic image of the Au/LaOx NPs/Pt device structure. The side shape of the LaOx NPs also was identified by the observations that the short and thick lines in Figure 1c have represented the layer thickness of the LaOx nanoplates. The ~1 µm of LaOx NP layer can be too thick for a high level of integration. However, the role of RS devices has been extended beyond an integrated platform.

To determine the optical band gap of the LaO_x_ NPs, the UV–vis spectrophotometer absorbance and Tauc plot were derived as shown in Figure 2. It was obtained by plotting the magnitude of light energy (*hv*) on the horizontal axis and the amount of light absorption (α*hν*)^r^ on the vertical axis. Here, α is the absorption coefficient and exponent *r* represents the nature of transition of the bandgap in the material. Since LaO_x_ NPs are in a polycrystalline state, they have a direct transition, and the exponent value (r) is 2 [23,25]. The band gap was found to be 4.98 eV, where a cross-point extends the tangent of the curve and meets the horizontal axis. Since most of the metal oxide’s optical bandgap is similar to the electrical bandgap, LaO_x_ electrical bandgap could be useful to estimate the electronic performance of the LaO_x_ NPs device.

The schematic illustration and photograph image in Figure 1 also show the deposited LaO_x_ NPs layer by convective self-assembled coating through the Van der Waals force and evaporation of the solution in meniscus. The area, having more evaporated time, showed a distinguishable color from the other area [26]. This phenomenon of polychromic layer formation was inevitable for the LaO_x_ NPs of this study, which were non-spherical or plate-shaped [25]. There has even been one report that the polychromic NP or nano-ribbon stacks have been characterized in terms of the RS effect with strontium titanate nanocubes [24]. However, this study is the first trial where the novel square-shaped NPs were adopted to fabricate a complete electronic device having top electrode [24,27].

As shown in Figure 1, the thickness of the grooved layer was measured at approximately 1 μm in thickness, particularly in the blue region. Appendix A shows a picture of completed the Au/LaO_x_ NPs/Pt device, and Appendix A shows the relative amount, which were calculated from the atomic ratios of La and O in the layer employing energy-dispersive spectroscopy (EDS) analysis. The EDS analyzed the amount of La and O, which were measured as small as 3.53 wt% and 5.96 wt%, respectively. However, it was seen that LaO_x_ NPs are well dispersed and distributed on the Pt surface. Since there should be less contacts out of Van der Waals binding between the Pt surface and the square-shaped NPs, the deposition process should be well tuned.

Figure 1 also shows a SEM image of the NP layer surface that resulted in the formation of repetitive groove/ridge patterns. The darker region of the dip-coated layer is the groove where the LaO_x_ NPs are deposited less. The lighter portion is the ridge where the LaO_x_ NPs stack more to form the ridge layer. On the meniscus between the air and Pt coated wafer, the LaO_x_ coating solution could generate the regular pattern because the stick–slip motion was caused by the force of pulling the substrate with a constant force and the surface tension of the LaO_x_ NP solution [24,26]. If spherical NPs were available and adopted, no ridge and groove pattern would be produced [3].

The RS characteristics of the Au/LaO_x_ NPs/Pt device were investigated by voltage sweep and pulse bias measurements. Figure 3a shows the current–voltage (I–V) curves obtained by a negative voltage sweep of 0 V → −7 V → 0 V and subsequent positive voltage sweep of 0 V → +7 V → 0 V, which was an excursive sweep (0 V → −7 V → 0 V → +7 V → 0 V) for successive nine times. In this case, it has been known that electronic–ionic interaction in the RE metal oxide films should play the key role for their RS functions [23]. For a comparison, I–V curves with a reference device of Al/LaO_x_ NPs/Pt device obtained by one positive voltage sweep of 0 V → −1.5 V → 0 V and subsequently negative voltage sweeps of 0 V → +1.5 V → 0 V for successive ten times in Appendix A. As shown in Appendix A, no apparent hysteresis similar to Figure 3a was observed with the Al top electrode.

Figure 3a shows asymmetric Schottky-type diode characteristics when the sweep voltage was excursively ranged from −7 V to +7 V. The rectifying behavior as shown in the positive sweep of Figure 3a is known to alleviate sneaky current in the RS devices [28,29].

During the first sweep of ±7 V, there were gradual increases in the magnitude of the current, which is known as the SET transition from a high resistance state (HRS) to a low resistance state (LRS). Nine consecutive excursive eight-wise sweeps (0 V → −7 V → 0 V → +7 V → 0 V) could demonstrate asymmetric RS characteristics with reproducible eight-wise polarity. However, the on/off ratio was small as it was less than 10^1^ under the compliance current, which was set to 10 mA [10]. During the excursive sweeps, the I–V curve was linear as the voltage was swept in a positive direction, implying Ohmic conduction through possibly generated conducting filaments (CFs). During repeated sweeps of 0 V → −7 V → 0 V → +7 V → 0 V, the SET transition occurred similarly in both positive and negative voltage regions. Severe change of I–V hysteresis was rarely found in all sweeps, which could form the corresponding hysteresis curve as an almost same shape, and any abrupt increase in the current was not observed during the switching from HRS to LRS. In the positive voltage sweep, the magnitude of hysteresis and current level were found to be much lower than the negative sweep. The reason for the difference in the positive and negative voltage regions could be due to the difference in the work function of the Pt bottom electrode and Au bottom electrode. There are several current irregularities in the 5th and 7th I–V sweep curves as shown in Figure 3a. They are believed to be from the square-shaped NPs’ edge effect with anisotropic contacts between individual LaO_x_ NPs.

Figure 3b shows a schematic illustration of charging at the LRS of the LaO_x_ NPs device. Since the work function of Au ranges from 5.37 to 5.47, while that of Pt ranges from 5.12 to 5.93, it can be easier to inject electrons from the Au electrode than the Pt bottom electrode as shown in Figure 3b [3,27,29]. Therefore, there could be difficulty in confirming the SET voltage that passed from the HRS to the LRS. Also, it indicates that negative voltage sweep (0 V → −7 V → 0 V) could have more efficient injection of electron through the Au top electrode to formulate a conductive path than the positive sweep.

Very often, theory about multiple RS has been resorted to conducting filaments (CFs), which is explained by formation through localized oxygen vacancies or oxygen ion (O^2−^) migration and electron hopping, which can induce the LRS [29,30]. However, in this study, the RS can be originated from charge trapping inside of LaO_x_ NPs and a series of conductive oxygen vacancy path on surface of LaO_x_ NPs stack due to the very thick LaO_x_ NPs layer (~1 μm). In addition, ruptures could have taken place along with continuous contacts among sandwiched individual LaO_x_ NPs.

Figure 3c shows I–V curves of extended hysteresis with decreasing resistance for consecutive voltage sweeps with increasing magnitude of ranges by ±3 V up to ±15 V. It was found that a decrease in the current level occurs for successive positive-voltage sweeps. Through subsequently repeating sweeps of negative and positive voltages by increasing −4 V and +4 V, the tendency in both directions remained the same, respectively, as shown in Figure 3c. Particularly, when the magnitude of the voltage sweep was increased by an absolute magnitude of 4 V, the current path in the negative sweep followed the previous history of the sweep path, whereas the positive voltage sweep did not follow the former trajectory of the positive voltage sweep at each voltage sweep. The adaptive evolvement in the negative sweep of sequential hysteresis demonstrates an analog memory effect, which can emulate a synaptic or neuromorphic functionality [1,29].

In addition, for I–V curves with a higher magnitude than ±5 V, self-rectifying functionalities were also demonstrated [28,29]. The threshold switching behavior can be very helpful in alleviating sneaky currents, which are available as a selection diode concurrent with the RS memory device [29,30]. The threshold RS effects having rectification functionality were also identified with a threshold voltage (V_th_) of ±2.5 V as shown in Figure 3c [29]. In addition, a forward-to-reverse current ratio of ~10^2^ could be demonstrated with a ±15 V sweep, as shown in Figure 3c [28].

Although the mechanism inducing the particular analog RS has not been perfectly clarified, it is expected that the analog RS was related to electrical charging in the NPs layer or minute redistribution of charged ions, such as O^2−^, at relative voltage within individual LaO_x_ NPs [2,3,4,5,24]. Therefore, the CFs which were induced by a series of individual LaO_x_ NPs could induce the analog switching behavior. The conduction mode analysis could disclose an apparent similarity that might come from asymmetric interfaces [20,29]. Typically, the threshold RS device has been characterized with its volatile property in its I–V characteristics [29].

The I–V curves in Figure 3a could roughly be fitted with Ohmic conduction or space–charge limited conduction (SCLC, I ∝ V*^n^*) as shown in Figure 4a,b. In this analog RS condition, the resistance was instantly changed during the voltage sweep. The appearance of Ohmic conduction (*n* = 0.94) at a transition from HRS to LRS in Figure 4 can be an evidence of CFs in the LaO_x_ NPs films [23].

Consequently, I–V curves were not fitted perfectly with a linear current form through the entire sweep voltage range. The SCLC mechanism (*n* = 2.85) is known to be influenced by the presence of space charges of particle surface, since the entire trap sites or charging nodes can be electrostatically occupied by charges and then current can be passed and facilitated through Coulombic repulsions between the charged traps and carriers [24,29]. It implies that the top Au/LaO_x_ and the bottom Pt/LaO_x_ interfaces could have different Schottky barriers. The barriers should be caused by the work function and the sequence of deposition process that the top Au electrode was finally deposited on the LaO_x_ NPs layer, while the bottom Pt electrode experienced the dip-coating process with LaO_x_. In Appendix A, the SCLC graphs for increasing voltage sweeps were chosen and specified. Even if the voltage sweep was gradually increased, it was fitted through the two individual conjunction mechanisms of SCLC.

The analog synaptic property was also examined as shown in Figure 5. The analog memristive behavior with respect to the polarity of voltage frequently acts like an adaptive synaptic motion of potentiation and depression-like biological neuro systems [24,29]. The depression having a gradual increase in resistance was confirmed by repeating +7 V pulse biases or spikes 10 times with 10 ms, 20 ms, 30 ms, 40 ms, 45 ms and 50 ms width time, respectively. Then, repeating a pulse of −7 V for 10 ms, 20 ms, 30 ms, 40 ms, 45 ms, 50 ms width ten times led to an increased current level of potentiation from 0.1 nA to 0.0172 A at the read voltage of +0.2 V. The potentiation behavior showed a typical long-term potentiation (LTP) effect with increased pulse time width. Particularly, when the −7 V pulse had a 40 ms width, it was found that the current or response due to stimuli or spikes was more sharply raised than in the shorter pulse width time (10 ms, 20 ms, and 30 ms). It is believed that the LaO_x_ NPs device clearly showed another example of synaptic functionality, i.e., neuron firing, with a longer stimulus duration than 30 ms. The neuron firing was assumed to be from a sudden rupture of the charge’s accumulations during the potentiation process due to an edge effect from the non-spherical shape of the LaO_x_ NPs [31]. In this case, the current increased from 0.016 nA to 0.102 nA as shown in Figure 5.

It was described that the loss rate would be reduced by an increasing number of pulses, with the degree of stimulus changing from short-term to long-term memory characteristics [31]. The loss rate would be determined by the number of charges and their energy levels in the traps of the NPs’ assembly structure.

As typically observed in the bipolar switching of a thick metal oxide NPs layer, applying positive voltage at the Au electrode resulted in SET transition during the interfacial reaction between NPs and the Au electrode, which would enrich metallic composition in the NPs layer [1,2,3,4,5,6,7].

Applying positive voltage at the Au electrode attracted O^2−^ ions to the Au electrode interface, which would be re-dissolved within the individual LaO_x_ NPs as an original reservoir of O^2−^ ions. As a result, oxygen vacancies or depletions inside of the LaO_x_ NPs were formed at the place where the O^2−^ ions had moved. In addition, several CFs might be formed by oxygen vacancies, which were believed to be created under the Au electrode. With the application of a negative voltage, the O^2−^ ions were migrated from the interface of the Au electrode into the LaO_x_ NPs. Therefore, RESET transition could occur where O^2−^ ions combine with the hole injected by positive bias. In this phenomenon, the presence of reactive RE LaO_x_ and even more existence of NPs itself is thought to facilitate the transitions due to a build-up of O^2−^ concentration by the applied electric field. Therefore, it is suggested that the CFs are barely formed and bridged by each LaO_x_ NP as shown in Figure 6. Each segment of filament is believed to be connected through grain boundaries formed by the LaO_x_ NPs’ edges at the smallest distance. Therefore, the CFs were believed to be formed in a relatively low voltage range even for the fast-electroforming process (<+7 V). In fact, a flat, thin film (50 nm) with a polycrystalline LaO_x_ phase was reported to have low set voltages (<2 V) for the RS device [23].

## 4. Conclusions

In this study, square-shaped LaO_x_ NPs with sizes of 20–30 nm were layered to consist dual functionalities of an analog RS non-volatile memory device and rectification diode. The stable I–V operations from a ±1 voltage double sweep to ±15 V double sweeps were attributed to bridged segments of CFs throughout the LaO_x_ layer induced under the electric field. Separately, a pulse bias of +7 V ranging from 10 ms to 50 ms width time showed LTP synaptic operations. The memristive device could demonstrate a potential applicability with non-spherical NPs for a synaptic neuromorphic device.

## Figures and Tables

**Figure 1 nanomaterials-11-00441-f001:**
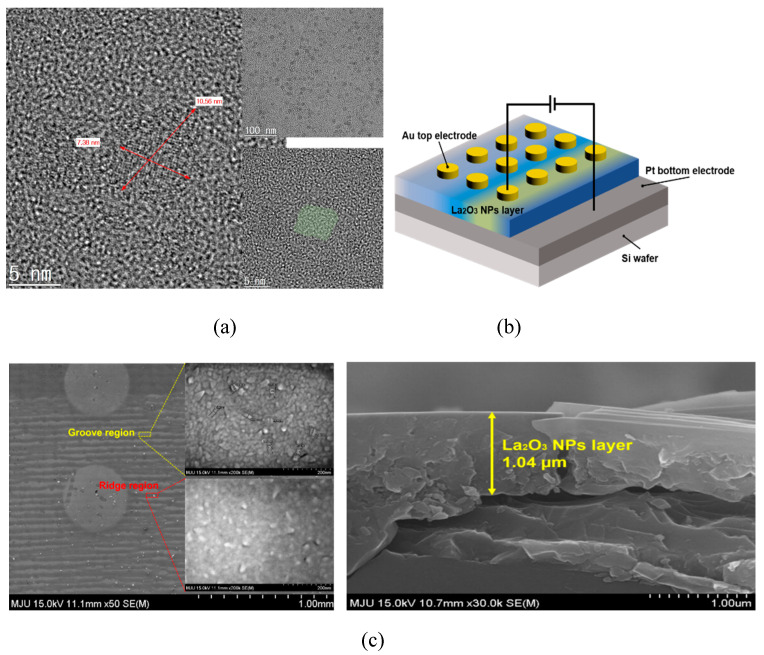
**(a)** TEM image of synthesized plate-shaped LaO_x_ NPs, (**b**) schematic image of the Au/LaO_x_ NPs/Pt device, (**c**) SEM image of groove/ridge pattern layer formed by LaO_x_ NPs after dip-coating, annealing and Au top electrode evaporation, groove and ridge pattern region on Pt coated wafer.

**Figure 2 nanomaterials-11-00441-f002:**
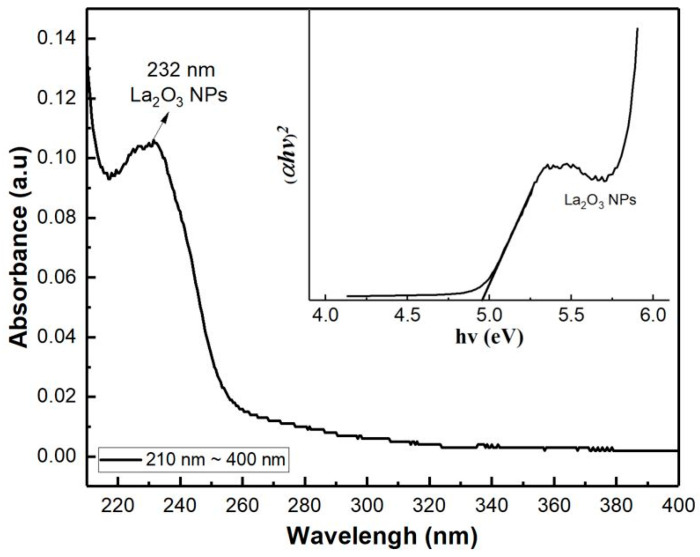
Absorbance spectra of UV–Vis spectrophotometer and band gap calculation using Tauc plot.

**Figure 3 nanomaterials-11-00441-f003:**
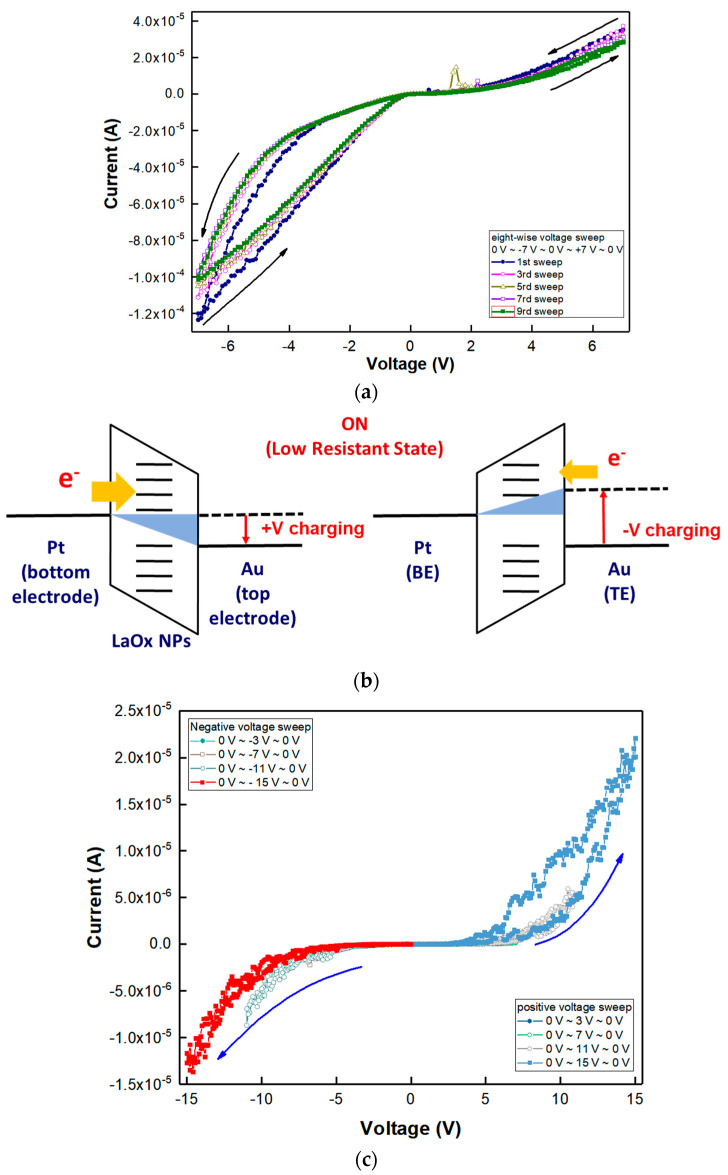
(**a**) Linear I-V plots for repeated excursive voltage sweeps of 0 V → −7 V → 0 V → 7 V → 0 V. (**b**) Schematic illustration of charging at a low resistance state (LRS). (**c**) Linear I–V curve for increasing voltage sweeps of 0 V → −3 V → 0 V to 0 V → −15 V → 0 V at −4 V increments and, subsequently, increasing voltage sweeps of 0 V → 3 V → 0 V to 0 V → 15 V → 0 V by increasing 4 V.

**Figure 4 nanomaterials-11-00441-f004:**
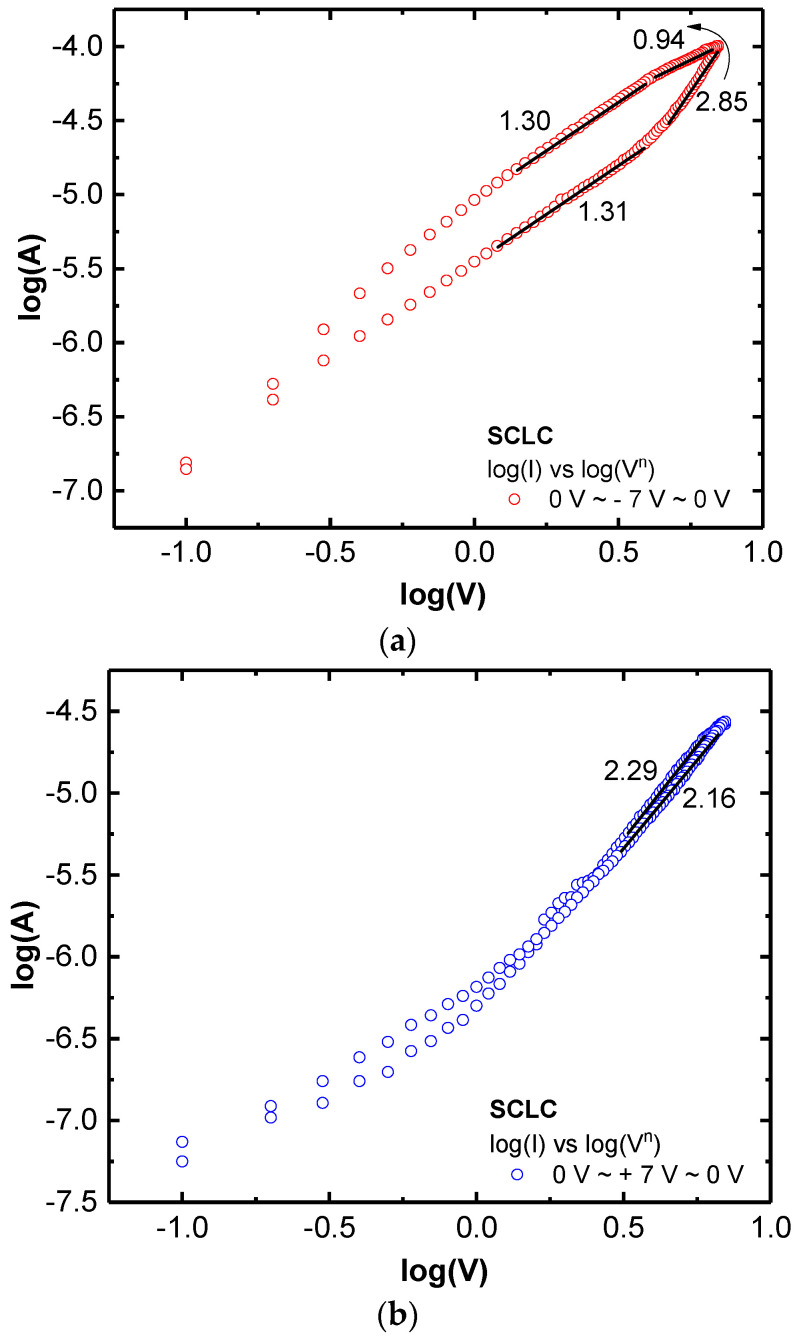
(**a**) Logarithmic plot for eight-wise voltage sweep (0 V → −7 V → 0 V → 7 V → 0 V) (**a**) plot of 0 V → −7 V → 0 V. (**b**) logarithmic plot for eight-wise voltage sweep (0 V → −7 V → 0 V → 7 V → 0.

**Figure 5 nanomaterials-11-00441-f005:**
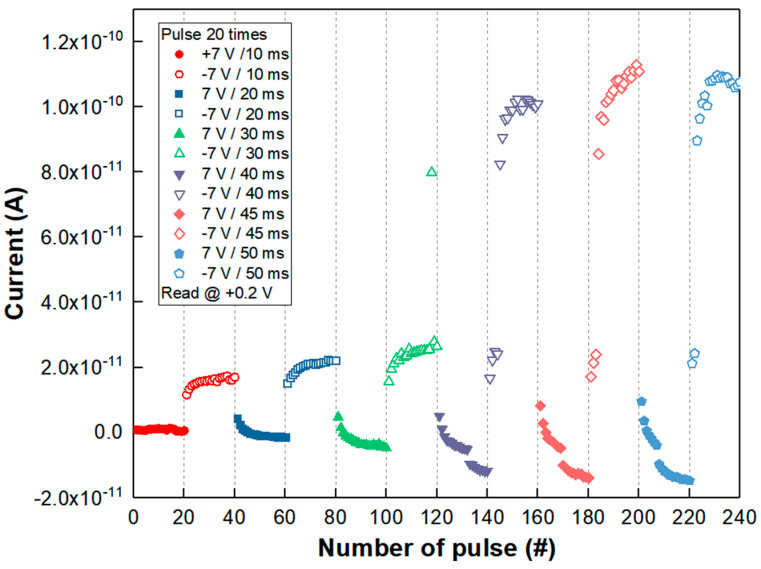
Current change of repeating pulses of +7 V for 10 ms, 20 ms, 30 ms, 40 ms, 45 ms, 50 ms ten times for depression and, subsequently, repeating a pulse of −7 V for 10 ms, 20 ms, 30 ms, 40 ms, 45 ms, 50 ms ten times for potentiation.

**Figure 6 nanomaterials-11-00441-f006:**
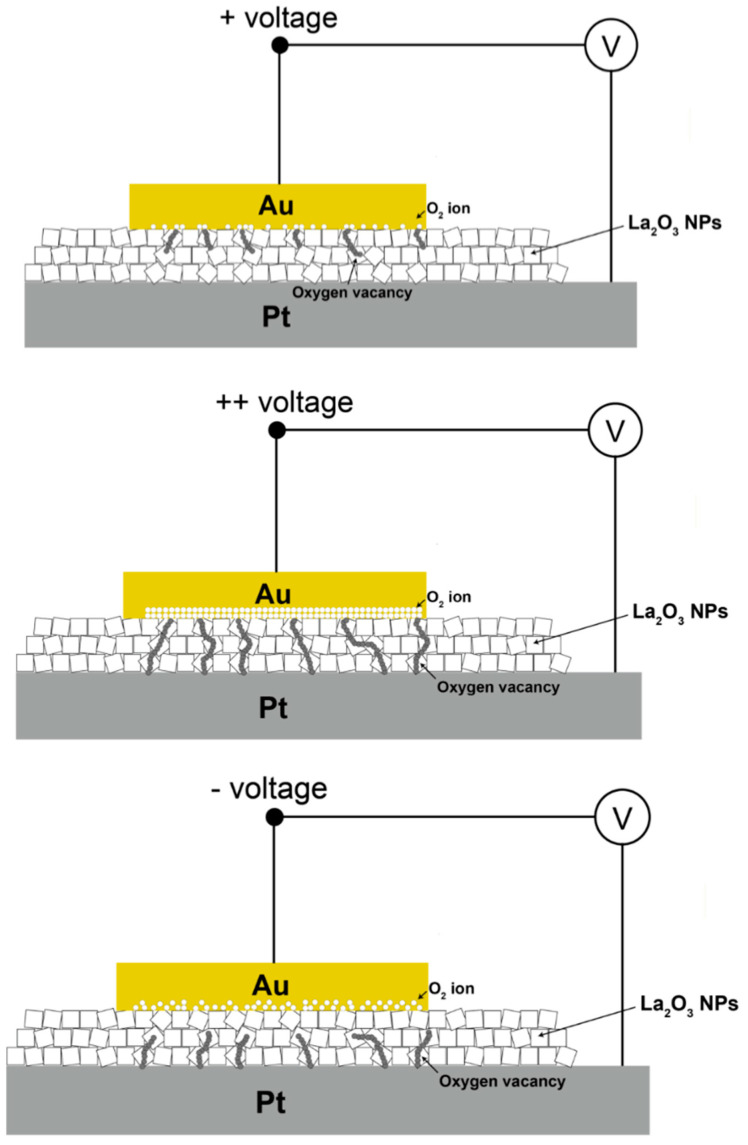
Schematic illustration of conduction filament formation in the mixed LaO_x_ NPs layer facilitated by bridged segments; initiation of filaments during the first SET; SET transition through filament formation at positive voltages, and RESET transition at negative voltages.

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
