# Peer review of "Analog Memristive Characteristics of Square Shaped Lanthanum Oxide Nanoplates Layered Device"

_nanomaterials, 2021, doi:10.3390/nano11020441_

Round 1

Reviewer 1 Report

Square-shaped or rectangular nanoparticles (NPs) of lanthanum oxide (LaOx) were synthesized and layered by convective self-assembly to demonstrate an analog memristive device in this study. This topic is interesting. Detailed comments and suggestions are given below. 1. It would be interesting to explain the reason why the current jumped up in some sweeps (1st 5th 7th sweep) in Figure 3a. Besides, too many curves are given in one figure (Figures 3a and 3c), which makes it unclear. It would be better to plot some of them. 2. It would be more clear to use different types of symbols to distinguish different curves in Figures 3 and 5.

Author Response

Square-shaped or rectangular nanoparticles (NPs) of lanthanum oxide (LaOx) were synthesized and layered by convective self-assembly to demonstrate an analog memristive device in this study. This topic is interesting. Detailed comments and suggestions are given below.

  1. It would be interesting to explain the reason why the current jumped up in some sweeps (1st 5th 7th sweep) in Figure 3(a).

Response: Authors would like to thank reviewer for valuable comments. There are several current irregularities in 5th and 7th I-V sweep curves as shown in Fig. 3(a). They are believed to be from square-shaped NPs’s edge effect with anisotropic contacts between individual LaOx NPs.

Besides, too many curves are given in one figure (Figures 3a and 3c), which makes it unclear. It would be better to plot some of them.

Response: Authors would like to thank reviewer for helpful comments. Figure 3(a) and 3(b) were simplified and re-drawn with less number of excursive sweeps.

  1. It would be clearer to use different types of symbols to distinguish different curves in Figures 3 and 5.

Response: Authors would like to thank reviewer for helpful comments. Figure 3 and 5 were re-drawn in distinguishable symbols.

Reviewer 2 Report

The manuscript reports on an experimental investigation of square-shaped or rectangular nanoparticles of lanthanum oxide, synthetized by chemical methods and investigated as analog memristive devices. The authors combine structural and morphological characterizations of the synthetized material with electrical I-V measurements. 

The novelty of the paper is related to the different fabrication approach used by the authors which results in improved robustness of the resistive-switching device based on lanthanum oxide. 

I believe that the manuscript may be of interest for the readers of Nanomaterials, but I suggest the authors to modify in part the language style to  better highlight the novel aspects of the manuscript. For example, I would use a more assertive style for the abstract where I would avoid the extensive use of forms like "could", e.g. "selection diode property could be consistent without any implementation of heterogeneous multiple...";....

Also I would suggest the authors to improve the readability of some figures, for example figure 3 is split between two pages, labelling of fig 3b is missing; figure 6 is labelled as "a" but it consists of a single panel (?).

Author Response

The manuscript reports on an experimental investigation of square-shaped or rectangular nanoparticles of lanthanum oxide, synthetized by chemical methods and investigated as analog memristive devices. The authors combine structural and morphological characterizations of the synthetized material with electrical I-V measurements.

The novelty of the paper is related to the different fabrication approach used by the authors which results in improved robustness of the resistive-switching device based on lanthanum oxide.

I believe that the manuscript may be of interest for the readers of Nanomaterials, but I suggest the authors to modify in part the language style to better highlight the novel aspects of the manuscript. For example, I would use a more assertive style for the abstract where I would avoid the extensive use of forms like "could", e.g. "selection diode property could be consistent without any implementation of heterogeneous multiple...";....

Response: Authors would like to thank reviewer for helpful language style comments. There are corrections for the assertive expressions from forms like “could”. At lines 18, 20, 21 of page 1, lines 50, 53, 68, 69, 70, 82, 89, 94, of page 2, lines 122, 141, 142 of page 3, lines of 170, 177, 178,  page 5, lines 280, 286, 292, of page 8, lines 298, 301, 303, 313, 319, 322 of page 9, lines 333, 349 of page 10, the forms in the expression of “could” were re-phrased with assertive ones.

Also I would suggest the authors to improve the readability of some figures, for example figure 3 is split between two pages, labelling of fig 3b is missing; figure 6 is labelled as "a" but it consists of a single panel (?).

Response: Authors would like to thank reviewer for typo comments. The labelling was added in Figure 3(b) in revised manuscript. And, wrong notation about Figure 6 was corrected by removing “(a)” notations.
